# Impact of Sex on Mortality in Patients Undergoing Surgical Aortic Valve Replacement

**DOI:** 10.3390/jpm12081203

**Published:** 2022-07-24

**Authors:** Hyun-Uk Kang, Jae-Sik Nam, Dongho Kim, Kyungmi Kim, Ji-Hyun Chin, In-Cheol Choi

**Affiliations:** Department of Anesthesiology and Pain Medicine, Asan Medical Center, University of Ulsan College of Medicine, Seoul 05505, Korea; hyunuk_kang@naver.com (H.-U.K.); jaesik_nam@naver.com (J.-S.N.); widrune94@gmail.com (D.K.); cjh@amc.seoul.kr (J.-H.C.); icchoi@amc.seoul.kr (I.-C.C.)

**Keywords:** surgical aortic valve replacement, aortic stenosis, survival analysis

## Abstract

Aortic stenosis (AS) is the second most common valvular heart disease in the United States. Although the prevalence of AS does not significantly differ between the sexes, there is some controversy on whether sex differences affect the long-term mortality of patients with severe AS undergoing surgical aortic valve replacement (SAVR). Therefore, we retrospectively analyzed the medical records of 917 patients (female, *n* = 424 [46.2%]) with severe AS who had undergone isolated SAVR at a tertiary care center between January 2005 and December 2018. During a median follow-up of 5.2 years, 74 (15.0%) male patients and 41 (9.7%) female patients died. The Kaplan–Meier analysis revealed that the 10-year mortality rate was significantly higher in male than female patients (24.7% vs. 17.9%, log-rank *p* = 0.005). In the sequential Cox proportional hazard regression model for assessing long-term mortality up to 10 years post-surgery, the adjusted hazard ratio of male sex for mortality was 1.93 (95% confidence interval, 1.28–2.91; *p* = 0.002). The association between male sex and postoperative long-term mortality was not significantly diminished by any demographic or clinical factor in subgroup analyses. In conclusion, female sex was significantly associated with better long-term survival in patients with severe AS undergoing SAVR.

## 1. Introduction

Aortic stenosis (AS) is currently the second most common valvular heart disease in the United States, where it affects about 12.4% of those aged over 75 years and has an increasing incidence with advancing age [1,2]. In a recent study, the proportion of people with AS was 80% in the population over 60 years of age in South Korea [3]. The pathological mechanism of AS is based on chronic pressure overload in the left ventricle (LV), which leads to changes in its geometry and molecular composition, including myocardial hypertrophy, ventricular dilation, and fibrosis [4]. Untreated symptomatic severe AS has a poor prognosis with a mortality rate of 50% within 2 years [5]. Thus, the American College of Cardiology/American Heart Association guidelines recommend that aortic valvular procedures be performed surgically or percutaneously in symptomatic severe AS cases, depending on the degree of surgical risk that can be taken [6].

Although the prevalence of AS does not significantly differ between males and females, there are sex differences in terms of valvular anatomy and hemodynamic adaptation to ventricular pressure overload [7,8]. Male hearts generally manifest myocardial fibrotic changes due to increased collagen synthesis and have a larger LV volume, mass, and mass/volume index than those of the female heart. In contrast, female hearts have more concentric hypertrophy of the LV, less ventricular dilation, and a smaller LV cavity than male hearts. Hence, females typically have a later onset of AS symptoms and a lower rate of referral [9].

It remains uncertain and controversial as to how sex differences affect the long-term mortality rate in severe AS patients undergoing surgical aortic valve replacement (SAVR). Some studies have reported that sex does not have a significant effect on survival outcomes of severe AS patients undergoing SAVR [10,11], whereas other studies have reported that female sex is associated with a poorer survival rate after SAVR [12]. Other studies have suggested that female sex is a significant predictor of better long-term survival due to predisposition and hormonal effects [13,14]. In the current study, we retrospectively evaluated the long-term mortality outcomes by sex in patients undergoing SAVR for AS and assessed other risk factors for survival.

## 2. Materials and Methods

### 2.1. Study Design and Patients

This study was an observational cohort study on patients with severe degenerative AS who underwent elective isolated SAVR at a tertiary care center (Asan Medical Center, Seoul, Korea) between January 2005 and December 2018. Using the institutional electronic medical records system, we identified all patients who had undergone scheduled SAVR for severe AS during the study period. Then, we excluded patients who had undergone previous transcatheter aortic valve replacement (TAVR), SAVR because of infective endocarditis, SAVR for rheumatic AS, concomitant moderate or severe aortic insufficiency, concomitant moderate or severe mitral stenosis or mitral regurgitation, concomitant hypertrophic cardiomyopathy, previous cardiac surgery, urgent surgery, combined coronary artery bypass graft surgery, a reoperation before discharge, and those who needed a percutaneous coronary intervention during the same hospitalization period.

This study was conducted according to the guidelines of the Declaration of Helsinki and was approved by the Institutional Review Board of Asan Medical Center (AMC IRB 2020-1582). This study is reported following the Strengthening the Reporting of Observational Studies in Epidemiology (STROBE) guidelines [15]. The need for written informed consent was waived due to the retrospective nature of this study. All clinical data were obtained from a retrospective review of the Asan Medical Center Cardiovascular Surgery and Anesthesia Database.

### 2.2. Preoperative Transthoracic Echocardiography

Comprehensive transthoracic echocardiography was conducted in all patients before elective SAVR by expert sonographers following the latest American Society of Echocardiography and European Association of Cardiovascular Imaging recommendations [16]. Two-dimensional images, M-mode, color flow Doppler images, and Doppler spectral analyses were obtained. The diagnosis and grading of AS were based on the transthoracic echocardiographic findings following the current guidelines [16]. Severe AS was defined as aortic valve area < 1.0 cm^2^, mean transaortic pressure gradient > 40 mmHg, and/or peak aortic jet velocity > 4 m/s.

Aortic valve areas were calculated using the continuity equation [16]. Relative wall thickness was defined as twice the LV posterior wall thickness divided by the LV diastolic diameter. Stroke volume was measured by Doppler echocardiography using the continuity equation. Arterial compliance was determined by the stroke volume-to-pulse pressure ratio and calculated using an equation described previously [17]. Valvuloarterial impedance was calculated as (systolic arterial pressure + transaortic mean gradient)/stroke volume index. The flow rate was calculated by the ratio of stroke volume to the ejection time, as described previously [18].

### 2.3. Outcome and Clinical Data

The outcome of this study was all-cause mortality after SAVR. The mortality data were obtained from the medical records and the National Health Insurance status. The end of the observational period was defined as either 10 years after the operation, the date of death, or 21 July 2020, whichever occurred first.

The baseline characteristics obtained from the study patients included age, body mass index (BMI), the European system for cardiac operative risk evaluation (EuroSCORE) II, preoperative electrocardiography results, history of systemic disease (i.e., congestive heart failure, cerebrovascular disease, peripheral vascular disease, or renal disease), medications, laboratory data, transthoracic echocardiographic values, type of prosthetic valve, and combined surgical procedures.

### 2.4. Statistical Analysis

Continuous data are presented as the mean ± standard deviation or the median (interquartile range), and categorical data are presented as numbers (percentages). Categorical data were compared using the chi-square test or Fisher’s exact test, as appropriate. Differences in long-term mortality by sex were compared using Kaplan–Meier curves and the log-rank test. The association between sex and long-term mortality was assessed using Cox proportional hazards regression analysis. The proportionality assumption was checked using rescaled Schoenfeld’s residuals over time, and there were no obvious violations. Univariate and multivariate Cox proportional hazard regression models were used to identify significant prognostic factors for long-term mortality.

Multivariate Cox proportional hazard regression models were sequentially constructed to examine the effect of demographic and clinical factors on long-term mortality in the study cohort, and were adjusted as follows: Model 1—age, BMI, and EuroSCORE II in addition to sex; Model 2—aortic valve cusp number (tricuspid aortic valve vs. non-tricuspid aortic valve) in addition to Model 1; Model 3— preoperative comorbidities (i.e., diabetes mellitus type 2 on insulin, coronary artery disease, congestive heart failure, cerebral vascular disease, chronic kidney disease, or chronic obstructive pulmonary disease) in addition to Model 2; and Model 4—calcium channel blocker medications in addition to Model 3. We also assessed any possible modifications of the associations between sex and other variables that were identified as risk factors for long-term mortality in univariate Cox proportional hazards regression analyses.

All reported two-sided *p*-values < 0.05 were considered significant. All statistical analyses were performed with R software version 3.6.3 (The R Foundation for Statistical Computing, Vienna, Austria).

## 3. Results

### 3.1. Baseline Characteristics According to Sex

A total of 1509 patients with severe AS underwent scheduled SAVR at our center during the study period. Of them, 917 patients were included in the final analysis (Figure 1). The baseline characteristics of the study patients are summarized according to sex in Table 1. Of the 917 patients, 424 (46.2%) were female. Compared with male patients, female patients were older (67.2 ± 9.4 vs. 65.3 ± 9.2 years; *p* = 0.003), had lower incidences of coronary artery disease and chronic obstructive pulmonary disease, had a higher EuroSCORE II and a lower hematocrit value, a higher proportion received angiotensin-converting enzyme inhibitors or angiotensin receptor blockers, and a lower proportion received beta blockers or clopidogrel.

The preoperative echocardiographic findings are summarized according to sex in Table 2. Compared with male patients, female patients had smaller LV dimensions, LV volume, LV mass index, and aortic valve area (0.57 ± 0.14 cm^2^ vs. 0.62 ± 0.14 cm^2^, *p* < 0.001). In contrast, female patients had a higher peak aortic jet velocity (5.25 ± 0.78 m/s vs. 5.11 ± 0.67 m/s, *p* = 0.002) and transaortic mean pressure gradient (64.47 ± 17.82 mmHg vs. 68.74 ± 21.80 mmHg, *p* < 0.001). Among female patients, 255 (60.1%) had concentric LV hypertrophy, while 42 (9.9%) had concentric LV remodeling. Among male patients, 268 (54.4%) had concentric LV hypertrophy, while 104 (21.1%) had concentric LV remodeling (*p* < 0.001).

### 3.2. Mortality after SAVR According to Sex

During a median follow-up period of 5.2 years, 115 (12.5%) patients died after SAVR, of whom 74 were male and 41 were female. The Kaplan–Meier analysis and log-rank test revealed that the postoperative 10-year mortality rate was significantly higher in male than female patients (24.7% vs. 17.9%, *p* = 0.005; Figure 2).

In the univariate Cox proportional hazard regression analysis (Figure 3, Table 3), the male sex was associated with a significantly higher risk of mortality after SAVR (hazard ratio [HR], 1.71; 95% confidence interval [CI], 1.17–2.50; *p* = 0.006). This association was consistent after adjusting for risk factors in multivariate analyses (Figure 3, Table 3). After adjusting for age, BMI, and the EuroSCORE II (Model 1), the adjusted HR of the male sex for mortality after SAVR was 2.12 (95% CI, 1.43–3.12; *p* < 0.001). The adjusted HR of male sex was 2.16 (95% CI, 1.46–3.20; *p* < 0.001) after adding the number of aortic valve cups (Model 2) and 1.82 (95% CI, 1.21–2.73; *p* = 0.004) after adding baseline comorbidities (Model 3). After further adjustment for calcium channel blocker medication (Model 4), the adjusted HR was 1.93 (95% CI, 1.28–2.91; *p* = 0.002).

### 3.3. Subgroup Analysis

Subgroup analysis conducted following sex stratification did not show significant interactions between sex and the other variables (i.e., age, BMI, EuroSCORE II, AV cusp number, congestive heart disease, cerebrovascular disease, chronic kidney disease, diabetes mellitus type 2 on insulin, or calcium channel blocker medication) that were identified as significant risk factors for long-term mortality in the univariate Cox proportional hazards regression analysis (Appendix A).

## 4. Discussion

Females with severe AS have been considered to have a higher risk of mortality after SAVR than males because of older age, additional comorbidities, and higher frailty at diagnosis [11,12]. Similarly, the female patients in our cohort were older, had higher EuroSCORE II scores, a low preoperative hematocrit, and higher proportions of concomitant diseases, such as hypertension, heart failure, and cerebrovascular disorders compared with male patients. Nevertheless, female sex was associated with significantly better long-term survival after SAVR in patients with severe AS.

Consistent with the results of previous reports [7,8,12], the preoperative echocardiographic findings in our cohort showed that females had smaller values of LV dimension, LV outflow tract diameter, and LV mass index, but higher values of peak aortic jet velocity and the transaortic mean pressure gradient. A previous study showed that the compensatory LV adaptation processes against LV pressure overload differ according to sex [7,8]. The hearts of females with AS display characterized responses to pressure loading concentric hypertrophy, such as maintaining a small LV cavity with normal systolic function and fast changes in collagen synthesis [7,19]. Further sex disparities have been found in the nitric oxide system and management of calcium and natriuretic peptides [20]. Hence, female hearts exhibit different morphological features, clinical symptoms, and disease consequences than male hearts.

Many studies have attempted to identify the possible effects of sex on postoperative outcomes from severe AS, but the issue remains controversial. Numerous reports have shown that female sex is associated with a poor prognosis following SAVR for severe AS [3,11,12,21,22,23]. However, most of those studies focused on patients who had undergone combined SAVR and coronary artery bypass graft surgeries [3,21,22] and found poorer short-term mortality outcomes in females. Previous studies have shown that females have a poorer long-term mortality outcome following coronary artery bypass graft surgery [24]. Contrary to these previous heterogeneous study populations, we evaluated AS cases that had solely undergone SAVR only, and only a few other studies have investigated the effect of sex on postoperative mortality in such an isolated SAVR cohort. Duncan et al. identified a higher in-hospital mortality rate in females who underwent isolated SAVR in an unadjusted analysis [11]; however, the adjusted short-term mortality outcomes were similar between the sexes. Moreover, unlike our study, Duncan et al. also included patients who had previously undergone cardiac surgery, had aortic insufficiency, a high grade of mitral regurgitation, or were emergent cases. Elhmidi et al. reported that female sex was an independent predictor of higher 30-day mortality risk in patients undergoing isolated SAVR, but not of long-term mortality [23]. However, that study included patients who underwent urgent and emergent operations and those who had endocarditis or underwent myomectomy. Chaker et al. also evaluated the differential effect of sex on outcomes from isolated SAVR, but did not find a significant association between sex and long-term mortality [12]. Those authors reported that only the in-hospital mortality rate was higher in females than males (3.3% vs. 2.9%). In contrast to our study, Chaker et al. included cases with previous sternotomy, those that had undergone concomitant procedures, and those with non-elective admission status.

The results of several other reports agree with our current observation that females have a better long-term survival rate than males after an isolated SAVR for AS [13,14]. Kulik et al. studied a comparable patient population to the current study and showed that females underwent fewer reoperations and had better long-term outcomes than males [13]. Fuch et al. observed similar short-term mortality outcomes between the sexes but a better long-term survival rate among females with advanced age [14]. Higgins et al. also evaluated the effect of sex on mortality in patients after SAVR [25] and reported that the 15-year survival rate after SAVR in females was superior to that in males. Another study suggested that these sex-specific differences in long-term mortality after SAVR might be attributable to differences in the general life expectancy and that female sex per se is an independent predictor of long-term survival after SAVR [26]. Moreover, one study showed that females tend to have better attendance rates at cardiac rehabilitation programs, which is also associated with reduced mortality [27].

We detected some differences between the preoperative baseline characteristics and the echocardiographic findings in our study population and those of previous studies. Fewer female patients in the current study had coronary artery disease or chronic obstructive pulmonary disease, and fewer underwent percutaneous coronary intervention compared with previous studies, even though the female patients undergoing SAVR in our series had a poorer preoperative condition than male patients due to their underlying baseline cardiovascular profile, socio-environmental factors, and access to care [7]. In addition, echocardiographic findings are typically worse in females than in males, but our results differed from previous reports in this respect [8]. The median ejection fraction and stroke volume index in our male patients were lower than those in female patients. The LV geometry in the current series was also different from that in previous reports, as only 10% of female patients and 20% of male patients showed concentric LV remodeling. The relative wall thickness was also higher in males than females in our cohort. These baseline discrepancies in the preoperative condition of the study patients may underlie the differences in the long-term outcomes between our study and previous studies.

TAVR is the preferred treatment modality for female patients and high-risk patients with severe AS. However, the Korean government controls the cost of health care through the National Health Insurance Service, and the indications for TAVR are narrower for Korean patients than for those in Western countries due to the cost of this procedure. Thus, some of our study patients undergoing SAVR may not have chosen to undergo TAVR even if it would have been a more suitable approach. Hence, even though TAVR is the first option for severe AS in high-risk patients, the clinical manifestations in patients undergoing SAVR in Korea range from mild symptoms to severely progressed disease. Our results may be a reflection of differences in the natural human lifespan between the sexes, as females live longer than males on average in most countries.

## 5. Study Limitations

Our current study had some limitations that should be discussed. First, the single-center retrospective nature of this study has the potential for selection bias. In addition, as the study population was almost entirely Korean and their treatments were controlled by the National Health Insurance Service, our data may simply reflect sex differences in lifespan. Additional multicenter studies are needed to verify the possible effects of the Korean health care system on our findings.

## 6. Conclusions

Although female AS patients in our study were older, had a higher EuroSCORE II, were more likely to have concomitant diseases and poor echocardiographic findings (i.e., smaller cavity size, smaller LV outflow tract diameter, and higher transaortic mean pressure gradient) compared with male patients, female sex was significantly associated with better long-term survival outcomes following SAVR for severe AS.

## Figures and Tables

**Figure 1 jpm-12-01203-f001:**
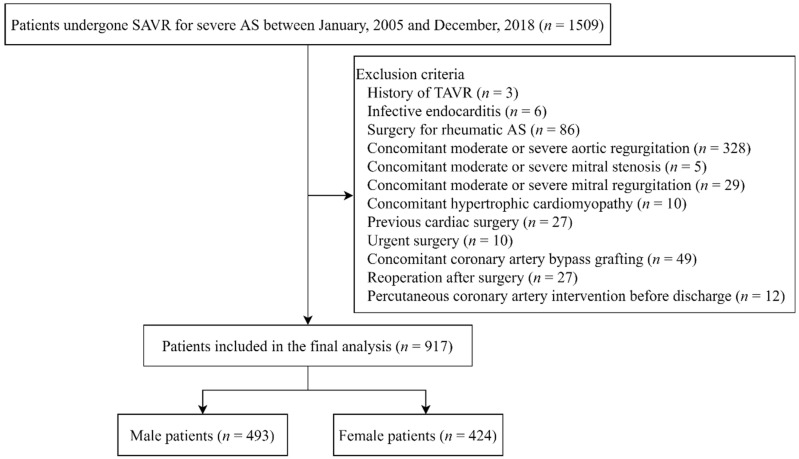
Study population flow chart. SAVR, surgical aortic valve replacement; AS, aortic stenosis; TAVR, transcatheter aortic valve replacement.

**Figure 2 jpm-12-01203-f002:**
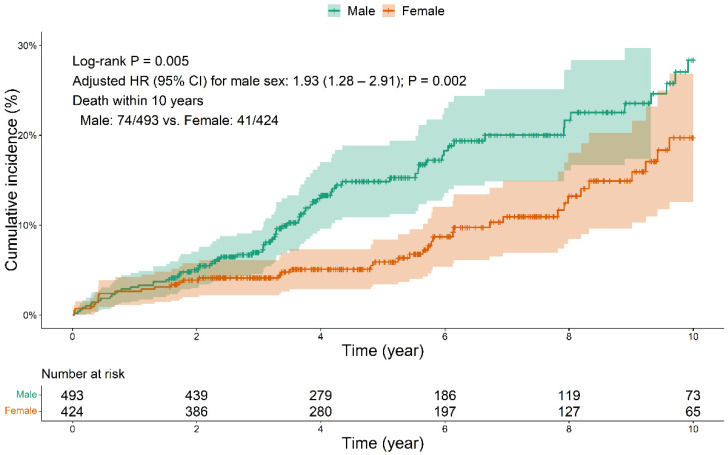
Ten-year mortality after surgical aortic valve replacement according to sex. HR, hazard ratio; CI, confidence interval.

**Figure 3 jpm-12-01203-f003:**
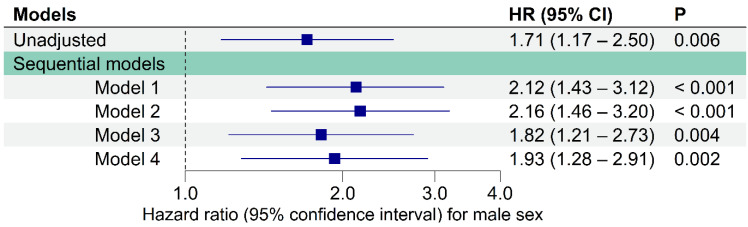
Unadjusted and adjusted hazard ratios for male sex in sequential multivariate Cox proportional hazards analyses of long-term mortality after surgical aortic valve replacement. The navy boxes in the plot indicate the hazard ratio of each model. Model 1: adjusted for age, BMI, and the EuroSCORE II as well as male sex. Model 2: adjusted for aortic valve cusp number (tricuspid aortic valve vs. non-tricuspid aortic valve) and Model 1. Model 3: adjusted for preoperative comorbidities (i.e., diabetes mellitus type 2 on insulin, coronary artery disease, congestive heart failure, cerebral vascular disease, chronic kidney disease, and chronic obstructive pulmonary disease) and Model 2. Model 4: adjusted for calcium channel blocker medications and Model 3. SAVR, surgical aortic valve replacement; HR, hazard ratio; CI, confidence interval; BMI, body mass index; EuroSCORE II, European system for cardiac operative risk evaluation II.

**Table 1 jpm-12-01203-t001:** Baseline characteristics according to sex.

	Male (*n* = 493)	Female (*n* = 424)	*p*
Age (years)	65.3 ± 9.2	67.2 ± 9.4	0.003
Body mass index (kg/m^2^)	24.7 ± 3.0	25.0 ± 3.4	0.132
EuroSCORE II	0.99 (0.74–1.70)	1.24 (0.89–2.02)	<0.001
Bioprosthetic aortic valve	218 (44.2%)	226 (53.3%)	0.020
Comorbidities			
Atrial fibrillation	26 (5.3%)	13 (3.1%)	0.099
Coronary artery disease	103 (20.9%)	49 (11.6%)	<0.001
Diabetes mellitus type 2	180 (36.5%)	130 (30.7%)	0.062
Hypertension	236 (47.9%)	227 (53.5%)	0.087
Myocardial infarction	7 (1.4%)	4 (0.9%)	0.559
Congestive heart failure	22 (4.5%)	21 (5.0%)	0.726
Percutaneous coronary intervention	41 (8.3%)	13 (3.1%)	0.001
Cerebrovascular disease	23 (4.7%)	25 (5.9%)	0.404
Peripheral vascular disease	25 (5.1%)	12 (2.8%)	0.086
Chronic obstructive pulmonary disease	35 (7.1%)	17 (4.0%)	0.044
Chronic kidney disease	38 (7.7%)	26 (6.1%)	0.350
Medications			
ACEi or ARB	150 (30.4%)	145 (39.2%)	0.031
Beta blocker	150 (30.4%)	102 (24.1%)	0.031
Calcium channel blocker	154 (31.2%)	153 (36.1%)	0.121
Insulin	83 (16.8%)	58 (13.7%)	0.186
Aspirin	157 (31.8%)	125 (29.5%)	0.439
Clopidogrel	64 (13.0%)	37 (8.7%)	0.040
Diuretics	145 (29.4%)	148 (34.9%)	0.075
Laboratory data			
Hematocrit	40.5 ± 4.6	36.7 ± 3.6	<0.001
Albumin	3.81 ± 0.40	3.80 ± 0.39	0.549
Estimated glomerular filtration rate	80.0 ± 19.4	82.3 ± 19.4	0.083
Brain natriuretic peptide	82.0 (39.0–210.3)	82.0 (36.5–240.0)	0.774

Values are the mean ± standard deviation, numbers (percentages), or the median (interquartile range). EuroSCORE II, European system for cardiac operative risk evaluation II; ACEi, angiotensin-converting enzyme inhibitor; ARB, angiotensin receptor blocker.

**Table 2 jpm-12-01203-t002:** Preoperative echocardiographic findings according to sex.

	Male (*n* = 493)	Female (*n* = 424)	*p*
LV end-diastole diameter (mm)	49.7 ± 5.9	46.7 ± 5.4	<0.001
LV end-diastole volume (mL)	110.0 (90.0–134.0)	84.0 (69.0–101.0)	<0.001
Relative wall thickness ^a^	0.49 ± 0.09	0.48 ± 0.09	0.300
Relative wall thickness < 0.42	121 (24.5%)	127 (30.0%)	0.066
LV ejection fraction (%)	63.0 (58.0–66.0)	64.0 (60.0–67.0)	<0.001
LV ejection fraction > 50%	436 (88.4%)	387 (91.3%)	0.158
LV mass (g)	239.7 ± 64.6	194.2 ± 53.3	<0.001
LV mass index (g/m^2^)	136.5 ± 35.5	125.8 ± 33.7	<0.001
LV geometry			<0.001
Normal	40 (8.1%)	42 (9.9%)	
Concentric remodeling	104 (21.1%)	42 (9.9%)	
Concentric hypertrophy	268 (54.4%)	255 (60.1%)	
Eccentric hypertrophy	81 (16.4%)	85 (20.0%)	
Number of aortic valve cusps			0.494
Unicusp	1 (0.2%)	1 (0.2%)	
Bicusp	297 (60.2%)	239 (56.4%)	
Tricusp	195 (39.6%)	184 (43.4%)	
Peak aortic jet velocity (m/s)	5.11 ± 0.67	5.25 ± 0.78	0.002
Transaortic mean pressure gradient (mmHg)	64.5 ± 17.8	68.7 ± 21.8	0.001
Aortic valve area (cm^2^)	0.62 ± 0.14	0.57 ± 0.14	<0.001
LV outflow tract diameter (mm)	21.4 ± 1.5	20.3 ± 1.3	<0.001
Stroke volume (mL)	75.1 ± 14.0	71.2 ± 13.4	<0.001
Stroke volume index (mL/m^2^)	42.9 ± 8.2	46.2 ± 9.1	<0.001
Stroke volume index < 35	79 (16.0%)	31 (7.3%)	<0.001
Transaortic flow rate (mL/s)	225.8 ± 40.8	211.6 ± 39.9	<0.001
Arterial compliance ^b^	0.67 ± 0.21	0.73 ± 0.22	<0.001
Valvuloarterial impedance ^c^	4.46 ± 1.02	4.27 ± 0.94	0.014

Values are the mean ± standard deviation, numbers (proportions), or the median (interquartile range). ^a^ Calculated as 2 × LV posterior wall thickness/LV diastolic diameter. ^b^ Calculated as stroke volume/pulse pressure. ^c^ Calculated as (systolic arterial pressure + transaortic mean pressure gradient)/stroke volume index. LV, left ventricle.

**Table 3 jpm-12-01203-t003:** Uni and multivariate Cox regression analyses of the effects of male sex on long-term mortality outcomes.

	Unadjusted		Model 1		Model 2		Model 3		Model 4	
	HR (95% CI)	*p*	HR (95% CI)	*p*	HR (95% CI)	*p*	HR (95% CI)	*p*	HR (95% CI)	*p*
Male sex	1.71 (1.17, 2.50)	0.006	2.12 (1.43, 3.12)	<0.001	2.16 (1.46, 3.20)	<0.001	1.82 (1.21, 2.73)	0.004	1.93 (1.28, 2.91)	0.002
Age			1.08 (1.05, 1.10)	<0.001	1.06 (1.03, 1.09)	<0.001	1.05 (1.03, 1.08)	<0.001	1.05 (1.03, 1.08)	<0.001
Body mass index			0.91 (0.86, 0.97)	0.002	0.89 (0.84, 0.95)	<0.001	0.88 (0.83, 0.94)	<0.001	0.88 (0.83, 0.94)	<0.001
EuroSCORE II			1.21 (1.05, 1.39)	0.009	1.26 (1.09, 1.45)	0.002	1.15 (0.98, 1.35)	0.082	1.18 (1.00, 1.38)	0.045
Tricuspid aortic valve					2.02 (1.32, 3.09)	0.001	1.69 (1.10, 2.60)	0.017	1.61 (1.04, 2.48)	0.032
Coronary artery disease							1.02 (0.64, 1.64)	0.931	0.94 (0.58, 1.52)	0.810
Congestive heart failure							1.36 (0.69, 2.68)	0.368	1.33 (0.68, 2.61)	0.413
Cerebrovascular disease							1.68 (0.94, 2.99)	0.077	1.61 (0.91, 2.88)	0.104
Chronic kidney disease							2.16 (1.33, 3.53)	0.002	1.98 (1.21, 3.25)	0.006
Chronic obstructive pulmonary disease							1.75 (0.96, 3.16)	0.065	1.68 (0.93, 3.06)	0.086
Diabetes mellitus type 2 on insulin							2.04 (1.28, 3.28)	0.003	1.97 (1.23, 3.17)	0.005
Calcium channel blocker medication									1.48 (1.00, 2.19)	0.051

Sequential modeling of the multivariate Cox proportional hazards analysis for long-term mortality was conducted using significant variables identified in the unadjusted Cox proportional hazards regression analysis for the male sex. The first model was unadjusted and included only sex, whereas subsequent models were adjusted as follows: [Model 1] age, BMI, and the EuroSCORE II in addition to sex; [Model 2] the aortic valve cusp number (tricuspid aortic valve vs. non-tricuspid aortic valve) in addition to Model 1; [Model 3] preoperative co-morbidities (i.e., diabetes mellitus on insulin, coronary artery disease, congestive heart failure, cerebral vascular disease, chronic kidney disease, and chronic obstructive pulmonary disease) in addition to Model 2; and [Model 4] calcium channel blocker medications in addition to Model 3. HR, hazard ratio; CI, confidence interval; EuroSCORE II, the European system for cardiac operative risk evaluation II.

## Data Availability

The data used in this study are available upon reasonable request from the corresponding author, but cannot be made publicly available due to the privacy of the patients.

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
