# Peer review of "Impact of Sex on Mortality in Patients Undergoing Surgical Aortic Valve Replacement"

_jpm, 2022, doi:10.3390/jpm12081203_

Round 1

Reviewer 1 Report

In the introduction, the authors describe the American data: "Aortic stenosis (AS) is currently the second most common valvular heart disease in the United States, where it affects about 12.4% of those aged over 75 years". Why do the authors not describe the disease data about the country they come from? The above sentence does not have to be deleted, but you can add another sentence with information about the disease statistics in your country. Yours observational study population was almost entirely Koreans - ethnically different from those living in the USA.

From many subgroups of patients, the group of patients was correctly separated for further analysis. Statistical analysis was very well conducted.

Author Response

July 21, 2022

Editor, the Journal of Personalized Medicine

Dear Dr. Dou,

Please find attached a revised version of our manuscript entitled “Impact of sex on mortality in patients undergoing surgical aortic valve replacement,” which we are resubmitting for publication in the Journal of Personalized Medicine.

Your comments and those of the reviewers were valuable and helpful for improving our manuscript. We have revised the manuscript following your recommendations, and our point-by-point responses to your comments have been provided separately.

We hope that our revisions have adequately addressed the reviewers’ concerns and that our manuscript is now acceptable for publication in the Journal of Personalized Medicine.

Yours sincerely,

Kyungmi Kim, M.D., Ph.D.

Clinical Assistant Professor

Department of Anesthesiology and Pain Medicine

Asan Medical Center, University of Ulsan College of Medicine

88, Olympic-ro 43-gil, Songpa-gu, Seoul 05505, Republic of Korea

E-mail: kyungmi_kim@amc.seoul.kr

Response to Reviewer #1 Comments

Point 1

In the introduction, the authors describe the American data: "Aortic stenosis (AS) is currently the second most common valvular heart disease in the United States, where it affects about 12.4% of those aged over 75 years". Why do the authors not describe the disease data about the country they come from? The above sentence does not have to be deleted, but you can add another sentence with information about the disease statistics in your country. Yours observational study population was almost entirely Koreans - ethnically different from those living in the USA.

Response to Point 1

We have added South Korean disease statistics information to the Introduction, as recommended (page 1, lines 27–28.

Related Revised Manuscript: Introduction

“Aortic stenosis (AS) is currently the second most common valvular heart disease in the United States, where it affects about 12.4% of those aged over 75 years and has an increasing incidence with advancing age [1,2]. In a recent study, the proportion of AS was 80% in the population more than 60 years of age in South Korea [3].

Point 2

From many subgroups of patients, the group of patients was correctly separated for further analysis. Statistical analysis was very well conducted.

Response to Point 2

We appreciate your time and effort on our manuscript (jpm-1818390).

Reviewer 2 Report

The subject of the work is very important. Aortic stenosis is one of the more serious problems. Surgical treatment and the approach to pharmacotherapy in this situation often requires individualisation.

In order to increase the value of your work, please make a few adjustments:

- mentioning diabetes as comorbidity, please specify which type. You can guess that it is DM vol.2 but please indicate it clearly.
- you mention that a certain percentage of patients with DM t. 2 were treated with insulin, what about the rest of the patients? What drugs were used.
- since the beginning of the 21st century, ACEi have been more and more boldly used in patients with aortic stenosis. Please comment on what may result in such a small percentage of ACEI use (e.g. in the discussion)
- The authors do not mention ARNI - please include in the introduction, for example
- what anticoagulation was used in both groups. Was she comparable. The type of anticoagulation may affect the mortality resulting from bleeding complications - please provide data in Table 1 and comment on these data
- was clopidogrel used in place of ASA or as a dual antiplatelet treatment together? Have other antiplatelet drugs like ticagrelor or prasugrel been used?
- what kind of diuretics? loop? HCTZ?
- is there anything known about physical activity?

After making corrections, I expect the work to be eligible for publication.

Author Response

July 21, 2022

Editor, the Journal of Personalized Medicine

Dear Dr. Dou,

Please find attached a revised version of our manuscript entitled “Impact of sex on mortality in patients undergoing surgical aortic valve replacement,” which we are resubmitting for publication in the Journal of Personalized Medicine.

Your comments and those of the reviewers were valuable and helpful for improving our manuscript. We have revised the manuscript following your recommendations, and our point-by-point responses to your comments have been provided separately.

We hope that our revisions have adequately addressed the reviewers’ concerns and that our manuscript is now acceptable for publication in the Journal of Personalized Medicine.

Yours sincerely,

Kyungmi Kim, M.D., Ph.D.

Clinical Assistant Professor

Department of Anesthesiology and Pain Medicine

Asan Medical Center, University of Ulsan College of Medicine

88, Olympic-ro 43-gil, Songpa-gu, Seoul 05505, Republic of Korea

E-mail: kyungmi_kim@amc.seoul.kr

Response to Reviewer #2 Comments

The subject of the work is very important. Aortic stenosis is one of the more serious problems. Surgical treatment and the approach to pharmacotherapy in this situation often requires individualisation.

In order to increase the value of your work, please make a few adjustments:

Point 1

mentioning diabetes as comorbidity, please specify which type. You can guess that it is DM vol.2 but please indicate it clearly.

Response to Point 1

All patients with diabetes mellitus in our study had type 2 DM. We have made this clear in Table 1 and indicate it on page 4.

Point 2

you mention that a certain percentage of patients with DM t. 2 were treated with insulin, what about the rest of the patients? What drugs were used.

Response to Point 2

A portion of the diabetic patients with or without insulin treatment had been taking oral hypoglycemics such as empagliflozin, metformin, linagliptin, or others. However, as insulin treatment implies worse glycemic control in type 2 DM patients, we considered that only insulin treatment among other glycemic control strategies could be a significant risk factor for outcome. Therefore, we decided not to present data other than insulin treatment. Additionally, as mentioned in the Study design and patients subsection in the Materials and methods, the study utilized clinical data extracted from a database separate from the institutional electronic medical records, which did not include the use of oral hypoglycemics or glycemic control with the diet.

Point 3

since the beginning of the 21st century, ACEi have been more and more boldly used in patients with aortic stenosis. Please comment on what may result in such a small percentage of ACEI use (e.g. in the discussion)

Response to Point 3

ACEi is a widely prescribed antihypertensive medication and has become one of the major components of medical treatment for aortic stenosis. A patient in South Korea who needs cardiac surgery may have consulted with a cardiologist after a short period of medical treatment, or may not have undergone a follow-up with any cardiologist before surgery. This point may have contributed to the low proportion of ACEi treatment. Furthermore, we evaluated the effect of differences in sex on mortality after surgical treatment for aortic stenosis, not the medical treatment. Although ACEi treatment is a cornerstone medical treatment option, we concluded that mentioning the effect of infrequent prescriptions for ACEi in the study population may be outside the range of the current study.

Point 4

The authors do not mention ARNI - please include in the introduction, for example

Response to Point 4

In the 2021 update to the 2017 ACC Expert Consensus Decision Pathway for Optimization for HF treatment, ARNI recommended ACEi or ARBs, highlighting the importance of this treatment option (J Am Coll Cardiol. 2021 Feb, 77 (6) 772–810). In South Korea, ARNI was first introduced to clinical practice in 2017. As we included patients who underwent SAVR from January 2005 to December 2018, the majority of patients underwent the surgery before ARNI was introduced to the country. Therefore, the use of ARNI before the surgery was not determined for the current study. Mentioning ARNI was outside the range of the current study.

Point 5

what anticoagulation was used in both groups. Was she comparable. The type of anticoagulation may affect the mortality resulting from bleeding complications - please provide data in Table 1 and comment on these data

Response to Point 5

We agree that the type of anticoagulation administered after valve replacement could have affected mortality from bleeding complications. However, as mentioned in the Response to Point 2, we utilized clinical data extracted from a database separate from the institutional electronic medical records. The database contained some information on the preoperative medications, but it did not contain any data relevant to postoperative medication usage. Hence, the information was not available to add to the current study.

Point 6

 was clopidogrel used in place of ASA or as a dual antiplatelet treatment together? Have other antiplatelet drugs like ticagrelor or prasugrel been used?

Point 7

what kind of diuretics? loop? HCTZ?

Response to Point 6 & Point 7

As mentioned in the Response to Point 2, we utilized clinical data extracted from a database separate from the institutional electronic medical records. Unfortunately, the only available data relevant to the patients’ preoperative medications were those listed in Table 1. Specifications of clopidogrel use as a single or dual antiplatelet treatment, use of antiplatelet agents other than aspirin or clopidogrel, and specifications of the kind of diuretic were not available.

Point 8

 is there anything known about physical activity?

Response to Point 8

The frequency and/or intensity of physical activity of each patient could have affected mortality after SAVR. Unfortunately, our database did not contain any data relevant to physical activity.

Reviewer 3 Report

In the paper "Impact of sex on mortality in patients undergoing surgical aortic valve replacement" the authors analyzed the effects of sex on mortality in patients who underwent surgical aortic valve replacement.

The research design is well conducted and the results clearly presented 

Minor point 

- Please revise the manuscript for typos/errors and to improve the language

- In order to better characterize the population add the mean dose and the SD of the drugs in Table 1 

-Please add a section on study limitations 

Author Response

July 21, 2022

Editor, the Journal of Personalized Medicine

Dear Dr. Dou,

Please find attached a revised version of our manuscript entitled “Impact of sex on mortality in patients undergoing surgical aortic valve replacement,” which we are resubmitting for publication in the Journal of Personalized Medicine.

Your comments and those of the reviewers were valuable and helpful for improving our manuscript. We have revised the manuscript following your recommendations, and our point-by-point responses to your comments have been provided separately.

We hope that our revisions have adequately addressed the reviewers’ concerns and that our manuscript is now acceptable for publication in the Journal of Personalized Medicine.

Yours sincerely,

Kyungmi Kim, M.D., Ph.D.

Clinical Assistant Professor

Department of Anesthesiology and Pain Medicine

Asan Medical Center, University of Ulsan College of Medicine

88, Olympic-ro 43-gil, Songpa-gu, Seoul 05505, Republic of Korea

E-mail: kyungmi_kim@amc.seoul.kr

Response to Reviewer #3 Comments

In the paper "Impact of sex on mortality in patients undergoing surgical aortic valve replacement" the authors analyzed the effects of sex on mortality in patients who underwent surgical aortic valve replacement.

The research design is well conducted and the results clearly presented 

Minor point 

Point 1

- Please revise the manuscript for typos/errors and to improve the language

Response to Point 1

The revised manuscript has been edited by at least two professional native English editors and we attached the editorial certificate.

Point 2

- In order to better characterize the population add the mean dose and the SD of the drugs in Table 1 

Response to Point 2

We agree that the specific dosage of each medication would have added information to the patient population. However, as mentioned in the Study design subsection of the Materials and methods, we utilized clinical data extracted from a database separate from the institutional electronic medical records. Unfortunately, the database did not contain specific label names or dosages of the medications administered preoperatively. Thus, these data were not available.

Point 3

-Please add a section on study limitations 

Response to Point 3

We have separated the Discussion section from the study limitations on page 9, lines 89-95.

Related Revised Manuscript:

5. Study limitations

Our current study had some limitations that should be discussed. First, the single-center retrospective nature of this study has the potential for selection bias. In addition, as the study population was almost entirely Korean and their treatments were controlled by the National Health Insurance Service, our data may simply reflect sex differences in lifespan. Additional multicenter studies are needed to verify the possible effects of the Korean health care system on our findings.

Round 2

Reviewer 2 Report

I suggest to accept without additional amendments